# A protocol for modeling the factors influencing the deployment of the COVID-19 vaccine across African countries

Obidimma Ezezika[1]*, Tiana Stephanie Kotsaftis[1], Edina Amponsah-Dacosta[2], Suleyman Demi[3], Eric Omori Omwenga[4], Samuel Mong'are[4], Trust Zaranyika[5], Oluwaseun Ariyo[6], Kandala Ngianga-Bakwin[7], Edward Kwabena Ameyaw[8]

**1** Faculty of Health Sciences, Global Health & Innovation Lab, School of Health Studies, University of Western Ontario, London, Canada, **2** Faculty of Health Sciences, Vaccines for Africa Initiative, School of Public Health, University of Cape Town, Cape Town, South Africa, **3** School of Social Work, Algoma University, Sault Ste Marie, Canada, **4** Department of Medical Microbiology and Parasitology, School of Health Sciences, Kisii University, Kisii, Kenya, **5** Department of Internal Medicine, University of Zimbabwe, Harare, Zimbabwe, **6** Department of Human Nutrition and Dietetics, College of Medicine, University of Ibadan, Ibadan, Nigeria, **7** Department of Epidemiology and Biostatistics, Schulich School of Medicine and Surgery, University of Western Ontario, London, Canada, **8** Institute of Policy Studies and School of Graduate Studies, Lingnan University, Tuen Mun, Hong Kong, China

* oezezika@uwo.ca

**Data Availability Statement:** All data are included in the manuscript.

## Abstract

Evidence fails to capture disparities amongst African countries in terms of the measure of administered COVID-19 vaccine doses per 100 people. Assessment of data on doses secured, administered, and supplied was undertaken to investigate quantitative measures that impacted COVID-19 vaccine deployment, thereby emphasizing distribution and supply indicators. We employ a full linear regression to identify independent variables that have an impact on vaccination rates, including macroeconomic indicators such as World Bank Income Classification, Gross Domestic Product (GDP) per capita and various indices such as Health Access and Quality Index, Human Development Index, Global Peace Index, Education Index, Political Stability Index and Government Effectiveness. This analysis aims to construct a statistical model utilizing regression analysis to identify key drivers of COVID-19 vaccine deployment in Africa and offer insights into vaccination disparities in the continent. Recognizing the global importance of achieving high vaccination rates, the study sheds light on specific challenges faced by individual countries within Africa, thereby emphasizing the need for tailored efforts. Beyond COVID-19, the research contributes to understanding the relationship between vaccination rates and social indicators that, potentially impact broader public health concerns and global vaccination programs. This study provides a foundation for informed policymaking to enhance vaccine accessibility, inform targeted programs, and improve individual health systems, thereby addressing broader implications for global health.

**Funding:** The author(s) received no specific funding for this work.

**Competing interests:** The authors have declared that no competing interests exist.

## Introduction

By the beginning of 2022, less than 10% of the African population was fully vaccinated against COVID-19. In contrast, half of the world was fully vaccinated at this time—77% and 63% of the population in countries such as Canada and the US, respectively, were fully vaccinated [1]. Vaccination rates for fully vaccinated individuals in Africa ranged from 80.7% in Seychelles and 0.1% in Burundi as of February 2022 [2]. Although other low-income countries and continents face or have previously faced COVID-19 vaccination concerns, Africa presents itself as a unique case. As highlighted above, the range of vaccination varies greatly along with their readiness for a COVID-19 vaccination program falling well below World Health Organization's (WHO) benchmark of 80% with 33% readiness [3]. This discrepancy extends beyond individual vaccination rates, as COVID-19 vaccine deployment has exhibited considerable variation across African countries, with only four Sub-Saharan African countries relatively smaller populations (Rwanda, Seychelles, Mauritius, and Cabo Verde) meeting the 70% vaccination target by mid-2022 [4].

Recognizing the need to better understand the underlying causes of these disparities, Statista provides comprehensive statistics on COVID-19 vaccine deployment, including the number of doses administered per 100 people, thereby enabling a quantitative rate assessment [5]. To obtain a better understanding of why such a disparity exists, it is worth exploring measures related to deployment to determine if there is a driving force or trend across African countries. For this purpose, COVID-19 vaccination rates, demographics, socioeconomic factors, social indexes/factors, and quantitative measures of access may be considered. Notably, socioeconomic factors—including World Bank Income Classification [6] and GDP per capita [7]—should be considered, as they may explain the disparities in vaccines secured and vaccination rates.

Moreover, previous literature has extensively delved into regression analyses on vaccination uptake and effectiveness, transcending the confines of COVID-19. The Centre for Disease Control (CDC) conducted a study utilizing regression and predictive modeling to compare vaccine coverage and uptake for influenza, pneumococcal disease, herpes, Td/Tdap, hepatitis A, hepatitis B, and HPV vaccines across the US based on demographic data [8]. These efforts expanded globally, with large-scale analyses examining general vaccine uptake and confidence across 149 countries from 2015 to 2019, thereby offering valuable insights into broader vaccination trends [9]. Further investigations have delved into the predictive and regression methodologies applied to specific vaccinations and social indicators. For example, Shaham et al. used predictive machine learning models with a focus on sociodemographic data to predict the rate of influenza vaccination in Israel [10]. Additionally, a study from Ghana focused on utilizing regression analysis to predict modifiable factors to increase HPV vaccination. uptake among adolescents such as beliefs about the efficacy of HPV vaccines [11].

There is a wide array of previous literature that has considered regression analyses in relation to vaccination effectiveness and impact on the total number of cases of COVID-19 inclusive of countries and areas across the world. Rustagi et al. considered vaccine data (first and second doses received) as well as the COVID-19 death ratio among 48 Asian countries to calculate the decrease in the number of cases and deaths [12]. An additional analysis focused on the COVID-19 infection rate in the US context [13] in which a statistical regression analysis was utilized to determine the effect of being vaccinated and staying home on daily incidence (13). In the European context, researchers utilized logistic regression models of vaccine hesitancy to determine what accounts for the variation in COVID-19 vaccine hesitancy across the continent [14]. Regression models were also used in determining the hospitalization risk associated with COVID-19 vaccines along with their effectiveness at reducing the rate of hospital

admissions in the United Arab Emirates [15]. Lastly, in a study across 13 Latin American and Caribbean countries, regression analysis was one of the methods used to determine the prevalence of the intention to be vaccinated against COVID-19 and associated factors [16].

Our proposed study will adopt a distinct approach by shifting focus from infection rates and vaccine efficacy and will address the research gap regarding the impact of social indicators on vaccination rates and uptake, specifically within the African context. With this perspective, predictive analyses will be employed to explore the influence of social indicators on COVID-19 dynamics. A comprehensive systematic review will investigate the global predictors of vaccine hesitancy and acceptance, utilizing regression models, with key findings highlighting perceived risk and institutional trust as predominant predictors of vaccine hesitancy [17].

Previous research has utilized social indicators to predict COVID-19 vaccination uptake on smaller scales, including Canada, Finland, and the US. Cheong et al. used machine learning analysis to predict COVID-19 vaccination uptake across various US counties by utilizing indicators such as location, education, income, etc. [18]. Statistics Canada conducted a similar study in the Canadian context to predict both vaccination intent and uptake [19]. Notably, Finland adopted a similar approach but introduced an innovative element by incorporating genome-wide information as a predictor to analyze the likelihood of COVID-19 vaccination uptake [20].

Thus, the impact of the social determinants of health is a well-established concept, but the examination of COVID-19 vaccination rates in the African context has remained limited. Despite minimal research, it has previously been identified that social factors such as high unemployment and low education can impact COVID-19 outcomes in Africa [21]. Recognizing this connection lays a foundation for further exploration of disparities and is crucial to understanding and addressing vaccination equity in Africa.

## Materials and methods

### Study aims

The overall aim is to understand the key predictors of the COVID-19 vaccination rate and the establishment of critical determinants that explain the variation in vaccination rates in Africa. Potential determinants and predictors are being ascertained from an ongoing systematic review on the upstream factors that might have impacted COVID-19 Vaccination Rates Across Africa [22].

Objectives of this study are as follows:

1. To examine COVID-19 vaccination disparities in Africa

2. To understand the ethical, social, cultural, historical, commercial, and logistical issues and factors that can explain the variation in COVID-19 vaccine implementation.

3. To develop an algorithm on COVID-19 country performance across supply, equitable distribution, demand, and financing activities to quantitatively assess scale–up efforts.

4. To explore the predictive impact of social indicators on COVID-19 vaccination rates.

### Data sources

We will use several data sourcessuch as Statista, Africa CDC, and COVAX to assess data on doses secured, administered, and supplied that have an impact on deployment. Although the sources for vaccines destroyed or expired are limited, doses secured [23] administered [5], and supplied [24] could all be considered to determine the number of vaccinations wasted. Doses

secured refers to how many COVID-19 vaccination doses each country received, whereas doses administered refers to how many doses were actually administered to individuals [5,23]. Vaccination rates and doses administered at different points in time [such as April 2022 [23] & March 2023 [5]] will be utilized to provide a comprehensive picture of the vaccination rate over time. Additionally, COVAX has separate data on how many donated COVID-19 vaccinations each country received [24], which will be utilized.

Based on the factors identified, an algorithm and regression analyses will be developed and applied to measure the performance of countries across key factors such as World Bank Income Classification [6], GDP per capita [7], Health Access and Quality Index [25], Human Development Index [26], Global Health Security Index [27], Global Peace Index [28], Education Index [29], Political Stability Index, government effectiveness, as well as voice and accountability measures [30]. Factors will be utilized to quantitatively assess each country's scale-up efforts and understand the gaps in deployment and vaccination rates that can uncover key predictors of vaccination rates, how they compare with each other, and critical insights and lessons for vaccine scale-up.

The Health Access and Quality Index [25], Human Development Index [21], Global Health Security Index [27], Global Peace Index [28], Education Index [29], Political Stability Index [30], government effectiveness [30], and voice and accountability measures [30] are all social factors that will be included. Considering a wide array of social indicators is essential to understanding thefactors that may influence countries' access or uptake of the COVID-19 vaccine from a social determinantsof health perspective. This could result in a better understanding of how efforts can be made to improve vaccine deployment.

## Hypotheses

We hypothesized that governments would be more likely to be successful in scaling up COVID-19 vaccination if they simultaneously, 1) made significant procurement of vaccines, either through public–private partnerships with African or other manufacturers; 2) conducted preparations in advance of vaccine delivery; 3) addressed infrastructure issues by scaling up resource mobilization, the availability of health posts, and the hiring and training of community and health care workers with the capacity to vaccinate, manage, and electronically track vaccinated individuals; 4) created strong access to quality health care as measured by the Healthcare Access and Quality (HAQ) Index [25], Human Development Index (HDI) Index [26], Global Health Security (GHS) Index [27], Global Peace Index [28], Political Stability Index, government effectiveness, and voice and accountability measures (30); 5) conducted sensitization and communication strategies to demystify the myths associated with COVID-19 vaccines and inform the population on vaccine safety and efficacy; 6) engaged champions and community-entrusted peers and advocacy groups to encourage the building of public trust and overcome vaccine hesitancy; 7) sought partnerships with external organizations—including Merck, Gavi, and others—to leverage foreign aid and international collaboration to support capacity building during initial COVID-19 vaccine rollout. In addition to governmental steps we hypothesized that income indicators (World Bank Income Classification and GDP per capita) will have a significant positive correlation to doses administered, secured, and supplied.

## Data analysis

Independent variables (doses supplied, doses secured, doses administered, doses wasted, population size, HAQ index, GDP per capita, and World Bank Income) and the dependent variable (COVID-19 vaccination rate) will be input to a log format Excel sheet aligned according to the

assigned country. To maintain consistency, we are standardizing the units with values in per capita being adjusted to the appropriate population size [31]. We are utilizing a reduced model and specifying it in an R data frame. The upstream independent factors stated previously, are being set as predictors in the model. These predictors will estimate the average population-level effect of each factor on the vaccination rate. Initially, a full linear regression model will be employed to determine if identified independent variables (Table 1) have an impact on vaccination rates across Africa. Subsequently, a model reduction approach will be applied to refine the model by selecting key indicators. The variables retained in the reduced model will be deemed pivotal factors impacting vaccination rates. The reduced model will then be compared to the full model using criteria such as adjusted R-squared and an ANOVA test to ensure that the simplification does not significantly compromise the model's explanatory power. Finally, the reduced model will be evaluated with the residual plot and the normal QQ plot to identify if the regression assumptions are satisfied. In cases where the regression assumptions are violated, alternative approaches such as robust regression will be considered to address any violations while maintaining the benefits of the reduced model.

## Pilot analyses

To illustrate, a preliminary reduced model was constructed to establish the influence of indicators (See Table 1) on vaccination rates. Six potential indicators—that is, independent variables —were utilized: population size, HAQ index, GDP per capita, doses administered, HDI index, and total area of the country. The final model, derived through our analysis, identifies two significant indicators—HAQ index and doses administered. Notably, the doses administered emerge as the most significant indicator, evidenced by a coefficient p-value of $<2e\text{-}16$. The positive coefficient associated with doses administered implies that higher number of doses administered correlates with an increase in vaccination rates. The adjusted R-squared of this model is 0.9981 and no sign of assumption violation was found. A detailed summary of the model is provided in the following Table 2.

## Model evaluation, strengths and limitations

To create a reduced model based on the correlation values, we aim to eliminate multicollinearity (high correlations) between predictors, to make our model more robust and interpretable.

**Table 1. Potential data sources for independent variables.**

| Data source | Reference |
| --- | --- |
| Doses Secured | Statista [23] |
| Doses Administered | Statista [5] |
| Doses Supplied | Africa CDC [24] |
| World Bank Income Classification | World Bank [6] |
| GDP Per Capita | World Bank [7] |
| Health Access and Quality Index | [25] |
| Human Development Index | United Nations [26] |
| Global Health Security Index | Global Health Security Index [27] |
| Global Peace Index | Institute for Economics & Peace [28] |
| Education Index | Rankedex [29] |
| Political Stability Index, government effectiveness, and voice and accountability measures | World Bank [30] |

**Table 2. Preliminary reduced model on factors influencing COVID-19 vaccination rates.**

| Term | Estimate | Std. Error | Statistic | P-Value |
|---|---|---|---|---|
| (Intercept) | -6.06475 | 1.26898 | -4.779 | 2.17e-05 |
| HAQ index | 0.07630 | 0.03628 | 2.103 | 0.0415 |
| Doses administered | 1.02590 | 0.01174 | 87.421 | < 2e-16 |
| GDP per capita v9 | -0.19627 | 0.16377 | -1.198 | 0.2375 |
| Population size v7 | 0.02090 | 0.02716 | 0.770 | 0.4458 |
| HDI index v11 | 2.68667 | 2.52514 | 1.064 | 0.2934 |
| (Intercept) | -6.06475 | 1.26898 | -4.779 | 2.17e-05 |

We will identify pairs of predictors with high absolute correlation values (e.g., > 0.7 or < -0.7) as these pairs indicate multicollinearity. We would then evaluate the adjusted $R^2$ values as a higher value indicates that the model is explaining a significant portion of the variance in the dependent variable, while also accounting for the number of predictors and avoiding overfitting. Once this is done, an ANOVA test will be run to identify significant differences in the models and see which reduced model significantly improves or worsens the model fit. For each pair of highly correlated predictors, we will decide which one to keep based on domain knowledge. We will select the reduced model that aligns with domain knowledge, has higher adjusted $R^2$ values, and performs best on the ANOVA test. To ensure model accuracy, we are conducting a comprehensive diagnostic analysis. We are using the plot() function to generate a set of four diagnostic plots. These plots include a residuals vs. fitted values plot to check for homoscedasticity and linearity assumptions, a Q-Q plot to assess normality of residuals, a scale-location plot to check for homogeneity of variance, and residuals vs. leverage plot to identify influential cases [32]. We will use Variance Inflation Factor (i.e. VIF) (model) to obtain the VIF values for the predictors in the final model to further examine the multicollinearity. A VIF value of a predictor larger than 10 will be regarded as high multicollinearity, and we may further investigate the corresponding predictor based on domain knowledge, to ensure robustness A summary of the model's performance across the five folds, including metrics such as RMSE and R-squared, will be provided to assess for overfitting to help us understand how well the model generalizes to new data.

## Ethics and dissemination

This protocol for modeling the factors influencing the deployment of the COVID-19 vaccine across African countries does not include patients or humans. The data generated through this modeling will be made fully available without restriction upon study completion.

## Discussion

A variety of factors affecting vaccine coverage across the continent have been documented, including (1) COVAX rollout, (2) vaccine storage and distribution capacity, (3) lessons learned from previous vaccination campaigns, (4) intellectual property, (5) vaccine procurement and (6) public reluctance. Despite this, the evidence fails to provide a holistic picture of factors that affect vaccine coverage across the continent.

A key initiative related to COVID-19 vaccine uptake among African countries was the creation of the COVAX initiative [33,34]. In 2020, COVAX was launched to ensure equitable global access to COVID-19 vaccines [28,29]. Despite its efforts to promote vaccine access for

low- and middle-income countries (LMICs), pressing demands for vaccines in the US, Europe, and other high-income nations shifted vaccine supplies, marketing, and deployment away from countries that needed this support [35,36]. There were severe allocation issues with COVAX facilities in delivering COVID-19 vaccines [34]. Moreover, certain African countries had inadequate capacity for vaccine storage and distribution. For example, although Sudan received the AstraZeneca vaccine through COVAX, accessibility challenges arose in the Darfur region due to the lack of vaccine storage and transportation facilities, shortages of healthcare professionals, and inequity in the distribution of health care facilities [37]. Despite several challenges, access was generally well facilitated by the different measures and initiatives adopted by African governments [38]. For example, Ghana's National Vaccine Deployment Plan (NVDP) was developed based on lessons learned during the yellow fever and polio immunization campaigns and was utilized to mobilize both human and logistical resources for COVID-19 vaccine deployment [39].

There were also several recurrent barriers to vaccine implementation around intellectual property, external policies, and procurement. Intellectual property (IP) rights for COVID-19 vaccines hindered the actualization of global vaccine access, significantly limiting vaccine accessibility and manufacturing capability in the African continent [36]. As a result of these barriers, African countries experienced the most significant shortages of COVID-19 vaccines compared to other regions [36]. Across 15 countries in West Africa, vaccinating the population was challenged by several external policies and circumstances. These included delays in obtaining emergency use authorization approvals from the World Health Organization (WHO)—a mechanism to facilitate the availability and utilization of medical countermeasures, including vaccines, during public health emergencies—and the halting of vaccine exports by the Serum Institute of India during the outbreak of COVID-19 in India [40]. Procurement of vaccines was a key hindrance due to differences in vaccine prices and availability between high-income and LMICs [41]. For example, the price per dose for AstraZeneca's vaccine was reportedly US$5.25 for South Africa, while European Union members were charged US$3.50 [41].

Existing literature has highlighted numerous reasons to why vaccination rates are low across the continent. Despite this, the evidence fails to provide a holistic picture of factors affecting vaccine coverage across the continent. Recent reflections on COVID-19 vaccine rollout can provide crucial insights for future pandemic preparedness and other biomedical technologies. Further, the study's potential reach and significance extend beyond geographical boundaries, and this study will have significant and far-reaching implications for overall pandemic preparedness and response against existing and emerging pathogens, including the rollout of other biomedical interventions such as antiviral therapies and rapid diagnosis tests.

While our study aims to analyze factors affecting COVID-19 vaccine deployment and uptake utilizing linear regression, addressing inherent limitations is necessary. Our study will consider all countries in Africa which have varying vaccination rates and supply which may leave outliers. The presence of outliers may lead to disproportionate influence affecting the data however they present it may be addressed through utilizing techniques to identify outliers and addressing them accordingly. In addition, the assumption of no multicollinearity, that independent variables are not correlated with each other may present a limitation, as many social determinants of health and factors are related to each other in some capacity. However, factors being utilized may still show that one has more of an influence than others and multicollinearity can be addressed through variance inflation factors and dimensionality reduction techniques if need be. Despite being inclusive of a wide array of data and the challenges that this brings, utilizing linear regression is crucial for highlighting significant relationships

between factors and vaccination rates to provide valuable insights into vaccine deployment and uptake in Africa.

Conducting this analysis will allow for strong upstream predictors to be highlighted which may be relevant to other regions. Countries with similar profiles may be able to utilize conclusions drawn to help guide efforts but relevance would need to be tested and confirmed with subsequent analysis. To elaborate, other low-income countries with similar factors may be able to use conclusions drawn if a country in Africa has similar scores on the variety of factors. Additional tests and replication of this study utilizing a different region is plausible but would involve further analysis.

Vaccines and their impacts are not exclusive to COVID-19. This study intends to fill a research gap and contribute to existing knowledge on COVID-19 vaccination in relation to social indicators. The research findings of the proposed study may have implications that extend to other public health concerns and global vaccination programs. The exploration of the social factors that influence COVID-19 vaccination rates can provide key insights and a foundation for informed policymaking to enhance vaccine uptake and accessibility. Vaccine uptake and accessibility are essential to safeguarding public health and wellbeing. Therefore, identifying the root causes ofbarriers can inform the scale-up of targeted vaccine programs. Moreover, it may help highlight indicators that, when improved, can positively impact the health care systems of individual countries.

## Author Contributions

**Conceptualization:** Obidimma Ezezika, Eric Omori Omwenga.

**Investigation:** Tiana Stephanie Kotsaftis, Edina Amponsah-Dacosta.

**Methodology:** Trust Zaranyika, Edward Kwabena Ameyaw.

**Supervision:** Obidimma Ezezika.

**Validation:** Edward Kwabena Ameyaw.

**Writing – original draft:** Obidimma Ezezika, Tiana Stephanie Kotsaftis, Trust Zaranyika, Kandala Ngianga-Bakwin.

**Writing – review & editing:** Tiana Stephanie Kotsaftis, Edina Amponsah-Dacosta, Suleyman Demi, Eric Omori Omwenga, Samuel Mong'are, Trust Zaranyika, Oluwaseun Ariyo, Kandala Ngianga-Bakwin, Edward Kwabena Ameyaw.

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
