## [Decision Letter · Decision Letter 0]

11 Jun 2024

PONE-D-24-09419A Protocol for Modeling the Factors Influencing the Deployment of the COVID-19 Vaccine Across African CountriesPLOS ONE

Dear Dr. Ezezika,

Thank you for submitting your manuscript to PLOS ONE. After careful consideration, we feel that it has merit but does not fully meet PLOS ONE’s publication criteria as it currently stands. Therefore, we invite you to submit a revised version of the manuscript that addresses the points raised during the review process.

We look forward to receiving your revised manuscript.

Kind regards,

Lipeng Song

Academic Editor

PLOS ONE

Journal Requirements:

Reviewers' comments:

Reviewer's Responses to Questions

**Comments to the Author**

1. Does the manuscript provide a valid rationale for the proposed study, with clearly identified and justified research questions?

Reviewer #1: Yes

Reviewer #2: Partly

2. Is the protocol technically sound and planned in a manner that will lead to a meaningful outcome and allow testing the stated hypotheses?

Reviewer #1: Yes

Reviewer #2: Partly

3. Is the methodology feasible and described in sufficient detail to allow the work to be replicable?

Reviewer #1: No

Reviewer #2: Yes

4. Have the authors described where all data underlying the findings will be made available when the study is complete?

Reviewer #1: Yes

Reviewer #2: Yes

5. Is the manuscript presented in an intelligible fashion and written in standard English?

Reviewer #1: Yes

Reviewer #2: Yes

6. Review Comments to the Author

You may also provide optional suggestions and comments to authors that they might find helpful in planning their study.

Reviewer #1: Authors apply the model to assess the influence of indicators on vaccination rates, which is very interesting. But, authors need to enhance the reliability of the method and the result need to There are the following points that need to intepret or supplement.

1. How to determine the candidate factor.

2. How to exclude pseudo-correlation.

3. Please interpret why to adopt linear regression model, and add the describe of the model and the research step of the stepwise regression approach.

Reviewer #2: The manuscript suggests a new mind to develop a holistic picture that may explain the variation in the COVID-19 vaccination rate both within and across African countries to highlight key predictors and strengthen pandemic responses in Africa. Employ algorithms and regression analyses are used to evaluate African countries’ performance across key factors. The specific comments are as follows:

1. There is a wide array of previous literature that has considered regression analyses in relation to vaccination effectiveness and impact on the total number of cases of COVID-19. Can we add literature to introduce the situation of other regions and countries, as Asian countries and the US are mentioned in the article?

2. There is very little introduction about the data analysis section. Can you provide a detailed introduction?

3. Can the conclusion drawn from the study of various African countries be compared with the results of other countries and regions?

4. Does the conclusion about African countries obtained in this article have reference value for vaccination efforts in other countries?

5. Overfitting and underfitting are also plausible limitations that linear regression is susceptible to. How was it specifically addressed in the paper?

6. Please use shorter sentences in the abstract section.

7. Please write in a standardized manner. For example, there is no punctuation at the end of the Hypotheses section.

7. PLOS authors have the option to publish the peer review history of their article (what does this mean?). If published, this will include your full peer review and any attached files.

Reviewer #1: No

Reviewer #2: No

---

## [Author Response · Author response to Decision Letter 0]

26 Jul 2024

Response to reviewers has been attached.

---

## [Editor Report · Decision Letter 1]

25 Sep 2024

A Protocol for Modeling the Factors Influencing the Deployment of the COVID-19 Vaccine Across African Countries

PONE-D-24-09419R1

Dear Dr. Obidimma Ezezika,

We’re pleased to inform you that your manuscript has been judged scientifically suitable for publication and will be formally accepted for publication once it meets all outstanding technical requirements.

Kind regards,

Osmond Ekwebelem

Academic Editor

PLOS ONE

---

## [Editor Report · Acceptance letter]

29 Oct 2024

PONE-D-24-09419R1 

PLOS ONE

Dear Dr. Ezezika, 

I'm pleased to inform you that your manuscript has been deemed suitable for publication in PLOS ONE. Congratulations! Your manuscript is now being handed over to our production team.

Kind regards, 

on behalf of

Dr. Osmond Ekwebelem 

Academic Editor

PLOS ONE